# Diagnosis of Latent Tuberculosis Infection in Hemodialysis Patients: TST versus T-SPOT.TB

**DOI:** 10.3390/diagnostics13142369

**Published:** 2023-07-14

**Authors:** Umut Devrim Binay, Ali Veysel Kara, Faruk Karakeçili, Orçun Barkay

**Affiliations:** 1Department of Infectious Diseases and Clinical Microbiology, Faculty of Medicine, Erzincan Binali Yıldırım University, 24100 Erzincan, Turkey; drfarukkarakecili@hotmail.com (F.K.); o.barkay1985@gmail.com (O.B.); 2Department of Nephrology, Faculty of Medicine, Erzincan Binali Yıldırım University, 24100 Erzincan, Turkey; aliveyselkara@hotmail.com

**Keywords:** hemodialysis, latent tuberculosis, TST, T-SPOT.TB

## Abstract

Hemodialysis (HD) patients should be screened for latent tuberculosis (TB) infection. We aimed to determine the frequency of latent TB infection in HD patients and to compare the effectiveness of the tests used. The files of 56 HD patients followed between 1 January 2021 and 1 October 2022 were retrospectively analyzed. Demographic data, the presence of the Bacillus Calmette-Guerin (BCG) vaccine, whether or not the patients had previously received treatment for TB before, the status of encountering a patient with active TB of patients over 18 years of age, without active tuberculosis and who had a T-SPOT.TB test or a Tuberculin Skin Test (TST) were obtained from the patient files. The presence of previous TB in a posterior–anterior (PA) chest X-ray was obtained by evaluating PA chest X-rays taken routinely. Of the patients, 60.7% (*n* = 34) were male and their mean age was 60.18 ± 14.85 years. The mean duration of dialysis was 6.43 ± 6.03 years, and 76.8% (*n* = 43) had 2 BCG scars. The T-SPOT.TB test was positive in 32.1% (*n* = 18). Only 20 patients (35.7%) had a TST and all had negative results. While the mean age of those with positive T-SPOT.TB results was higher (*p* = 0.003), the time taken to enter HD was shorter (*p* = 0.029). T-SPOT.TB test positivity was higher in the group that had encountered active TB patients (*p* = 0.033). However, no significant difference was found between T-SPOT.TB results according to BCG vaccine, albumin, urea and lymphocyte levels. Although T-SPOT.TB test positivity was higher in patients with a previous TB finding in a PA chest X-ray, there was no statistically significant difference (*p* = 0.093). The applicability of the TST in the diagnosis of latent TB infection in HD patients is difficult and it is likely to give false-negative results. The T-SPOT.TB test is not affected by the BCG vaccine and immunosuppression. Therefore, using the T-SPOT.TB test would be a more appropriate and practical approach in the diagnosis of latent TB in HD patients.

## 1. Introduction

Tuberculosis (TB) continues to be one of the main causes of death due to infectious diseases all over the world. The World Health Organization (WHO) has implemented the ‘End tuberculosis’ strategy and in relation to this, it recommends screening and treating latent TB infection (LTBI) [1]. According to the Turkish Ministry of Health’s Tuberculosis Diagnosis and Treatment Guidelines, it is recommended that patients with a high risk of latent TB reactivation, such as hemodialysis (HD) patients, should be screened. Since the risk of transmission will be high in hemodialysis units, the development of tuberculosis disease in this patient group must be prevented [2]. When prior studies are examined, in the systematic reviews conducted by Alemu et al., LTBI and active tuberculosis infection were found to be more common in dialysis patients [3,4]. In the study of Xia et al., it was found that the rate of development of active tuberculosis was higher in hemodialysis patients with LTBI. In the same study, in which patients were followed up about three years, LTBI was also shown to be associated with major adverse cardiovascular events [5]. In the study of Park et al., it was shown that active tuberculosis is more common in dialysis patients and kidney transplant recipients compared to the general population and causes higher mortality rates [6]. In the study of Romanowski et al., it was found that active tuberculosis was seen less frequently in patients who were treated for LTBI [7].

Interferon Gamma Release Assays (IGRA) and the Tuberculin Skin Test (TST) are used in LTBI screening, and it is recommended that IGRA should be performed in immunocompromised groups such as hemodialysis patients when the TST is negative or cannot be performed. Among the IGRA tests, the T-SPOT.TB, QuantiFERON-TB Gold In Tube (QFT-GIT) or QuantiFERON-TB gold plus test are used [2,8,9,10,11,12]. When studies comparing the diagnostic tests are examined, there is no gold standard test. In the study of Akbar et al., the QuantiFERON-TB gold plus test was shown to be superior to the TST. However, the small sample size was determined as a limitation of the study [13]. On the other hand, in the study of Setyawati et al., the use of the TST was recommended in the diagnosis of LTBI [14]. However, although it is stated that IGRA tests are not affected by immunosuppression, studies in patients with chronic kidney disease have shown that, as the time of dialysis increases, IGRA tests are more likely to give false-negative results. In this context, it is recommended that patients with chronic kidney disease be screened for LTBI at an early stage [15,16,17,18]. Considering the systematic reviews carried out in recent years, it has been shown that IGRA tests are superior to the TST [11,19,20].

The aim of this study is to determine the frequency of latent TB infection in patients undergoing hemodialysis in our hospital and to compare the effectiveness of tests used in the diagnosis of latent TB infection. At the same time, the aim is to investigate the reasons for the inconsistency between the tests by determining the factors affecting the tests.

## 2. Materials and Methods

### 2.1. Study Design and Population

This study was planned as a retrospective, cross-sectional study and was conducted with the approval of Erzincan Binali Yildirim University Clinical Research Ethics Committee (Date: 10 November 2022/Decision No: 05/09).

Latent TB infection screening is routinely performed in the hemodialysis unit of our institution. Patients are regularly referred to a tuberculosis dispensary for a TST to be performed. The T-SPOT.TB test is performed simultaneously with a hemogram, biochemical examinations and a posterior–anterior (PA) chest X-ray taken during routine dialysis, for patients who cannot undergo a TST or whose results are negative.

So, the files of HD patients who were followed up in the hemodialysis unit of a tertiary research and training hospital between 1 January 2021 and 1 October 2022 were reviewed retrospectively. Demographic data of the patients (age, gender, comorbidities, duration of hemodialysis admission, etc.) and data on the presence of Bacillus Calmette-Guerin (BCG) vaccine, whether or not they had previously received treatment for active TB, and their prior encounters with a patient with active TB were obtained from patient files. The presence of previous TB in a PA chest X-ray was obtained by evaluating the PA chest X-rays taken routinely.

The inclusion criteria were:To have regular hemodialysis;To be over 18 years old;To have T-SPOT.TB or TST results;To not have a concurrent active TB diagnosis.

Accordingly, out of a total of 67 patients, 56 patients who met the inclusion criteria were included in the study. Since the number of patients who did not undergo a TST was high, a comparison of both tests could not be made. Therefore, the factors affecting the results of the T-SPOT.TB test and the TST were evaluated separately.

### 2.2. Methodology

After blood samples were taken using special tubes for the T-SPOT.TB test (Oxford Immunotec, Oxford, UK), T-Cell Xtend reagent was added to the blood samples and sent to the laboratory. Mononuclear cells were obtained by centrifugation from the blood taken for the T-SPOT.TB test. The resulting mononuclear cells were added to wells previously coated with IFN-γ antibodies. Then, the TB antigens ESAT-6 and CFP-10 and Phytohemagglutinin were added for positive control. The negative control was determined as the well without antigens. These wells were incubated overnight at 37 °C with 5% CO_2_. After incubation, the wells were washed and secondary conjugated antibodies were added to measure the IFN-γ response. Spots that formed in the wells in which the IFN-γ response was observed were measured by an automated ELISPOT reader (AID systems, Strassberg, Germany). The result was considered positive if the test wells contained at least five more spot-forming cells than the average of the negative control wells [21].

The TST was applied intradermally to the upper inner 2/3 of the left forearm of the patients, in a hairless area away from the veins, with an insulin injector, with 0.1 mL of 5 TU PPD containing tuberculin solution. The transverse diameter of the formed induration was measured in mm after 48–72 h. Results with an induration diameter of 5 mm or more were considered positive [22].

The hemogram test of the patients was performed using the Sysmex XN-1000 Hematology System (Sysmex Corporation, Kobe, Japan); biochemical tests were performed with AU 5800 (Beckman Coulter, Brea, CA, USA).

### 2.3. Statistical Analysis

The NCSS (Number Cruncher Statistical System) 2007 (NCSS LLC, Kaysville, UT, USA) program was used for statistical analysis. Descriptive statistical methods (mean, standard deviation, median, frequency, ratio, minimum, maximum) were used while evaluating the study data. The conformity of quantitative data to normal distribution, the Shapiro–Wilk test and graphical evaluations were used.

Student’s *t*-test was used for comparisons of normally distributed quantitative variables between two groups, and the Mann–Whitney U test was used for comparisons of non-normally distributed variables.

Pearson’s chi-squared test, the Fisher–Freeman–Halton test and Fisher’s exact test were used to compare qualitative data. Logistic regression analysis was used in multivariate evaluations of the risk factors affecting T-SPOT.TB positivity.

Significance was evaluated at the *p* < 0.05 level.

## 3. Results

The study was carried out at a research and training hospital between 1 January 2021 and 1 October 2022. It was carried out with 56 HD patients, of whom 39.3% (*n* = 22) were female and 60.7% (*n* = 34) were male. The ages of the patients ranged from 20 to 81, with a mean of 60.18 ± 14.85 years. The duration of HD ranged from 1 to 27 years, with a mean of 6.43 ± 6.03 years. In total, 66.1% (*n* = 37) of the cases had comorbidities. When the types of comorbidities were examined, it was observed that 32.4% (*n* = 12) had type 2 Diabetes Mellitus (DM), 78.4% (*n* = 29) had essential hypertension (HT) and 45.9% (*n* = 17) had other diseases.

Of the patients, 3.6% (*n* = 2) had a prior history of active TB. The patients stated that they had completed their treatment. The number of patients who had encountered active tuberculosis patients was 8.9% (*n* = 5).

In addition, 8.9% (*n* = 5) of the patients had no BCG vaccine scar, 14.3% (*n* = 8) had one scar and 76.8% (*n* = 43) had two scars (Table 1).

Mean leukocyte counts were 6992.86 ± 3842.15/mm^3^; mean lymphocyte counts were 1981.43 ± 3533.68/mm^3^; mean albumin level was 17.42 ± 16.16 g/dL; and mean urea level was 149.18 ± 28.91 mg/dL.

Overall, 7.1% (*n* = 4) of the patients had previous tuberculosis findings on a chest X-ray.

The T-SPOT.TB test results were negative in 67.9% (*n* = 38) and positive in 32.1% (*n* = 18). A TST was performed in 35.7% (*n* = 20) of the patients and all of them were negative (Table 1).

### 3.1. Assessment of T-SPOT.TB Results

A statistically significant correlation was found between age and the T-SPOT.TB test result (*p* = 0.003; *p* < 0.01). The mean age of the group with positive results was found to be higher than the group with negative results.

A statistically significant correlation was found between the time on HD and the T-SPOT.TB test result (*p* = 0.029; *p* < 0.05). The HD time in the group with positive results was found to be shorter than in the group with negative results.

While the T-SPOT.TB test results did not show a statistically significant difference by gender (*p* = 0.072; *p* > 0.05), it is noteworthy that the rate of positive results in men was higher than that in women.

There was no statistically significant correlation between the presence of comorbidities and the T-SPOT.TB test results (*p* > 0.05).

A statistically significant correlation was found between encountering a patient with active tuberculosis and the T-SPOT.TB test results (*p* = 0.033; *p* < 0.05). The rate of positive results in the group that encountered a tuberculosis patient was higher than the group that had not.

No statistically significant correlation was found between the number of BCG scars and the T-SPOT.TB test results (*p* > 0.05) (Table 2).

No statistically significant correlation was found between leukocyte count, lymphocyte count, albumin level and urea level and the T-SPOT.TB test results (*p* > 0.05).

While no statistically significant correlation was found between previous TB findings on PA chest X-rays and the T-SPOT.TB test results (*p* = 0.093; *p* > 0.05), it is noteworthy that the rate of positive results was higher in the group with a previous TB finding in a PA chest X-ray (Table 3).

When we evaluated the risk factors affecting the T-SPOT.TB test, such as age, gender, a previous TB finding on a PA chest X-ray, time of dialysis and encountering an active tuberculosis patient with Enter Logistic Regression Analysis, the model was found to be significant and the explanatory coefficient of the model (76.8%) was found to be at a good level. It is seen that the effect of a unit increase in age on T-SPOT.TB positivity increases the ODDS ratio 1.101 (95% CI: 1.016–1.192) times. The effect of encountering an active tuberculosis patient has an effect on T-SPOT.TB positivity with an ODDS value of 59.762 (95% CI:1.59–2233.42) times. The effects of gender, a previous TB finding on a PA chest X-ray and time of dialysis were not significant in the multivariate evaluation (*p* > 0.05) (Table 4).

### 3.2. Results of Patients with Known History of TST

A TST was performed in 35.7% (*n* = 20) of the patients and it was found that all of them had negative results. Of these cases, 30% (*n* = 6) were female and 70% (*n* = 14) were male. Their ages ranged from 20 to 81 years, with a mean of 61.85 ± 17.1 years. The duration of HD ranged from 1 to 20 years, with a mean of 6.40 ± 6.08 years.

In total, 50% (*n* = 10) of the cases who underwent a TST had comorbidities. When the types of comorbidities were examined, it was observed that 20% (*n* = 2) had type 2 DM, 90% (*n* = 9) had essential hypertension and 50% (*n* = 5) had other diseases.

The rate of having had tuberculosis previously was found to be 5% (*n* = 1), while 20% (*n* = 4) of the patients who underwent a TST stated that they had previously encountered an active tuberculosis patient.

When the BCG scars of the tested participants were examined, 10% (*n* = 2) had no scar, 15% (*n* = 3) had one scar and 75% (*n* = 15) had two scars (Table 5).

Mean leukocyte counts were 7000 ± 4336.87 mm3; mean lymphocyte counts were 2209 ± 3951.76 mm^3^; the mean albumin level was 16.39 ± 16.41 g/dL; and the mean urea level was calculated as 141.95 ± 22.97 mg/dL. There was no significant difference between leukocyte and lymphocyte count according to the positive and negative T-SPOT.TB test.

In 10% (*n* = 2) of the patients who underwent a TST, previous TB was found in a PA chest X-ray.

The T-SPOT.TB test results of the patients who underwent a TST and all had negative results were found to be 20% (*n* = 4) negative and 80% (*n* = 16) positive (Table 6).

## 4. Discussion

The frequency of latent TB infection and the probability of TB reactivation in hemodialysis patients are higher than in the normal population [1,2,23,24,25]. Therefore, HD patients should be screened for latent TB infection [1,2]. The TST and IGRA are used in screening, and in a study investigating the frequency of latent TB infection in HD patients in low- and high-risk groups for latent TB infection, IGRA were shown to be superior to the TST [26]. In our study, the rate of latent TB infection was 32.1% and was similar to the study of Bandiara et al. that was conducted in HD patients (39.1%) [27]. In another study conducted in Thailand, the frequency of latent TB infection in dialysis patients was found to be 25% [28]. In the study of Lemrabott et al., 25% of latent TB infection was found in Senegal [29]. Similar rates of latent TB infection were found in the study of Putri et al. [17]. The frequency of latent TB infection in our study was as high as in Rheumatoid Arthritis and Ankylosing Spondylitis patients in the other immunosuppressive patient group [30].

The rate of latent TB infection in our study could be determined with the T-SPOT.TB test because only 35.7% (20) of the patients had a TST and all of them had negative results. Most of the patients (64.3%) refused to go to the tuberculosis dispensary for a TST. This shows that the applicability of the TST in hemodialysis patients is difficult. At the same time, in the study of Say et al., in which the QFT-GIT and TST were compared for the diagnosis of latent TB infection, there was no concordance between the two tests [31]. In the study of Southern et al., there was a high degree of discordance between IGRA and the TST in hemodialysis patients [15]. In a study conducted with HIV-infected individuals, another group of immunosuppressive patients, moderate concordance was found between T-SPOT.TB and the TST, and it was stated that the discordance might be due to false-positive and -negative results of the TST [32]. The disadvantages of the test are that the TST gives false positive results in the presence of the BCG vaccine and atypical mycobacterial infection, and false-negative results in the presence of immunosuppression [33,34]. In our study, T-SPOT.TB positivity was found in 16 (80%) of 20 patients whose TST results were negative. This result shows that the TST gives false-negative results. On the other hand, the expensiveness of IGRA is the disadvantage of the tests, while the advantages are that they are not affected by the presence of the BCG vaccine, atypical mycobacterial infection and immunosuppression [34,35]. In the study of Sargın et al., which was conducted in a rheumatologic patient group, the sensitivity and specificity of IGRA tests were shown to be superior to the TST [36]. In our study, no significant correlation was found between T-SPOT.TB test results according to BCG vaccine status, and this shows that the test is not affected by the BCG vaccine. The BCG vaccine is in the routine childhood vaccination schedule in our country and is administered at the end of the 2nd month [2]. It is important for our country that the T-SPOT.TB test is not affected by the BCG vaccine. Although the number of patients is small, the positive T-SPOT.TB test in patients with a TB finding on a PA chest X-ray indicates that the probability of a false-negative result is low. However, false-negative results should be investigated in HD patients, including in many patients with a history of microbiologically proven tuberculosis.

In our study, T-SPOT.TB test positivity was statistically higher in patients who had encountered active TB patients. In the study of Park et al., QFT GIT positivity, which is one of the IGRA tests, was higher in HD patients who had a history of TB [37]. In a study, T-SPOT.TB test positivity in patients with high risk factors, such as encountering an active TB patient, shows that the probability of developing active TB is higher [38]. In this case, it is important to initiate latent TB treatment without delay in high-risk patients with a positive T-SPOT.TB test.

It has been shown that advanced age, active smoking and close contact with someone who has previously had TB are among the risk factors for latent TB infection in HD patients. In the same study, it was stated that high albumin levels and short HD duration facilitate the detection of latent TB infection [39]. In a study conducted in Japan in hemodialysis patients, the frequency of LTBI was found to be higher, especially in people aged 60 and over [40] and in other studies conducted in China and Lebanon, advanced age was found among the risk factors for latent TB infection [5,41]. In our study, the high mean age of the patients with a positive T-SPOT.TB test and the shorter time to enter HD in patients with a positive T-SPOT.TB test support the literature. The incidence of TB in our country has been decreasing over the years, so that the incidence of TB, which was 29.8% in 2005, decreased to 14.4% in 2018 [42]. This may be the reason why the T-SPOT.TB test gives high positive results in older age groups. In our study, there was no significant difference between positive/negative results in terms of albumin, urea and lymphocyte levels, while the average albumin levels of patients with a T-SPOT.TB positive result were higher. However, the fact that there was no significant correlation between the T-SPOT.TB test results according to the urea levels and lymphocyte counts of the patients suggests that the test is not affected by immunosuppression.

The small number of patients and the fact that many patients did not have a TST are the limitations of the study.

In conclusion, HD patients should be screened for latent TB infection as soon as possible. Although it is recommended to perform a TST first in screening, the applicability of the test is not easy and the possibility of false-negative results is high, which limits its use. The most important advantage of the T-SPOT.TB test is that it is not affected by immunosuppression and it is studied with a single measurement from blood. Therefore, the use of the T-SPOT.TB test would be a more practical and accurate approach to screen for latent TB infection in HD patients.

## Figures and Tables

**Table 1 diagnostics-13-02369-t001:** Demographic Characteristics and Distribution of Laboratory and Imaging Results of the Patients.

		Min-Max (Median)	Mean ± Sd
Age (year)		20–81 (63)	60.18 ± 14.85
Time of dialysis (years)		1–27 (4)	6.43 ± 6.03
		*n*	%
Gender	Female	22	39.3
	Male	34	60.7
Presence of Comorbidity	Yes	37	66.1
	No	19	33.9
Type of comorbidities	Type 2 DM	12	32.4
	Hypertension	29	78.4
	Other *	17	45.9
History of active tuberculosis	Yes	2	3.6
	No	54	96.4
Encounter with active tuberculosis patient	Yes	5	8.9
	No	51	91.1
BCG scar	0	5	8.9
	1	8	14.3
	2	43	76.8
Leukocyte count (mm^3^)		2800–25,400 (6250)	6992.86 ± 3842.15
Lymphocyte count (mm^3^)		440–21,020 (1185)	1981.43 ± 3533.68
Albumin (g/dL)		2.1–41 (3.8)	17.42 ± 16.16
Urea (mg/dL)		85–238 (147.5)	149.18 ± 28.91
		*n*	%
Previous TB finding on PA chest X-ray	Yes	4	7.1
	No	52	92.9
T-SPOT.TB test	Negative	38	67.9
	Positive	18	32.1
TST	Negative	20	35.7
	Unknown	36	64.3

* Chronic Obstructive Pulmonary Disease, Alport Syndrome, Chronic Lymphoproliferative Leukemia, Coronary Artery Disease, Rheumatoid Arthritis.

**Table 2 diagnostics-13-02369-t002:** Relationship between Descriptive Characteristics and T-SPOT.TB Results.

		T-SPOT.TB Test		*p*
		Negative (*n* = 38)	Positive (*n* = 18)	
Age (year)	Min-Max (Median)	20–80 (57)	38–81 (71)	^a^ 0.003 *
	Mean ± Sd	56.26 ± 15.08	68.44 ± 10.6	
Time of dialysis (year)	Min-Max (Median)	1–27 (5)	1–20 (3)	^b^ 0.029 *
	Mean ± Sd	7.45 ± 6.37	4.28 ± 4.71	
Gender	Female	18 (81.8)	4 (18.2)	^c^ 0.072
	Male	20 (58.8)	14 (41.2)	
Presence of Comorbidity	Yes	27 (73)	10 (27)	^c^ 0.253
	No	11 (57.9)	8 (42.1)	
Type of comorbidities (*n* = 37)				
Type 2 DM	Yes	8 (66.7)	4 (33.3)	^d^ 0.696
	No	19 (76)	6 (24)	
Essential HT	Yes	20 (69)	9 (31)	^d^ 0.404
	No	7 (87.5)	1 (12.5)	
Other *	Yes	13 (76.5)	4 (23.5)	^d^ 0.725
	No	14 (70)	6 (30)	
Encounter with active tuberculosis patient	Yes	1 (20)	4 (80)	^d^ 0.033 *
	No	37 (72.5)	14 (27.5)	
BCG scar	0	3 (60)	2 (40)	^e^ 1.000
	1	6 (75)	2 (25)	
	2	29 (67.4)	14 (32.6)	

^a^ Student’s *t*-Test; ^b^ Mann–Whitney U Test; ^c^ Pearson’s Chi-Squared Test; ^d^ Fisher’s Exact Test; ^e^ Fisher–Freeman–Halton Test; * Chronic Obstructive Pulmonary Disease, Alport Syndrome, Chronic Lymphoproliferative Leukemia, Coronary Artery Disease, Rheumatoid Arthritis.

**Table 3 diagnostics-13-02369-t003:** Relationship between Laboratory and Imaging Results and T-SPOT.TB Test Results.

		T-SPOT.TB		*p*
		Negative (*n* = 38)	Positive (*n* = 18)	
Leukocyte (mm^3^)	Min-Max (Median)	2800–13,300 (6350)	4000–25,400 (5400)	^b^ 0.352
	Mean ± Sd	6602.63 ± 2108.09	7816.67 ± 6085.11	
Lymphocyte (mm^3^)	Min-Max (Median)	440–2710 (1165)	660–21,020 (1275)	^b^ 0.516
	Mean ± Sd	1322.11 ± 562.14	3373.33 ± 6057.06	
Albumin (g/dL)	Min-Max (Median)	2.9–41 (3.8)	2.1–39 (17.9)	^b^ 0.853
	Mean ± Sd	16.46 ± 16.06	19.46 ± 16.65	
Urea (mg/dL)	Min-Max (Median)	85–238 (152)	103–180 (142.5)	^a^ 0.230
	Mean ± Sd	152.39 ± 31.85	142.39 ± 20.61	
Previous TB finding on PA chest X-ray	Yes	1 (25)	3 (75)	^d^ 0.093
	No	37 (71.2)	15 (28.8)	

^a^ Student’s *t*-Test; ^b^ Mann–Whitney U Test; ^d^ Fisher’s Exact Test.

**Table 4 diagnostics-13-02369-t004:** Logistic Regression Results of Factors Affecting T-SPOT.TB Test.

	*p*	ODDS	95% C.I.ODDS	
			Lower	Upper
Age	0.018 *	1.101	1.016	1.192
Gender (F)	0.128	3.937	0.674	23.003
Previous TB finding on PA chest X-ray (+)	0.311	3.766	0.290	48.857
Time of dialysis (year)	0.827	1.017	0.875	1.182
Encounter with active tuberculosis patient (+)	0.027 *	59.762	1.599	2233.422

* *p* < 0.05,

**Table 5 diagnostics-13-02369-t005:** Distribution of the Descriptive Characteristics of the Patients who had TST.

*N* = 20		Min-Max (Median)	Mean ± Sd
Age (year)		20–81 (64)	61.85 ± 17.10
Time of dialysis (year)		1–20 (3.5)	6.40 ± 6.08
		n	%
Gender	Female	6	30
	Male	14	70
Presence of Comorbidity	Yes	10	50
	No	10	50
Type of comorbidities (n = 10)	Type 2 DM	2	20
	HT	9	90
	Other *	5	50
History of active tuberculosis	Yes	1	5
	No	19	95
Encounter with active tuberculosis patient	Yes	4	20
	No	16	80
BCG scar	0	2	10
	1	3	15
	2	15	75

* Chronic Obstructive Pulmonary Disease, Alport Syndrome, Chronic Lymphoproliferative Leukemia, Coronary Artery Disease, Rheumatoid Arthritis.

**Table 6 diagnostics-13-02369-t006:** Distribution of Laboratory and Imaging Results of the Patients who had TST.

*N* = 20		Min-Max (Median)	Mean ± Sd
Leukocyte (mm^3^)		2800–21,900 (5400)	7000 ± 4336.87
Lymphocyte (mm^3^)		660–18,850 (1285)	2209 ± 3951.76
Albumin (g/dL)		2.1–39 (3.7)	16.39 ± 16.41
Urea (mg/dL)		100–180 (142.5)	141.95 ± 22.97
		n	%
Previous TB finding on PA chest X-ray	Yes	2	10
	No	18	90
T-SPOT.TB test	Negative	4	20
	Positive	16	80
	T-SPOT.TB test		*p*
	Negative (n = 4)	Positive (n = 16)	
Leukocyte (mm^3^)	2800–13,300 (6350)	4200–21,900(5400)	^b^ 0.892
	7175.0 ± 4593.7	6956.3 ± 4426.4	
Lymphocyte (mm^3^)	800–2210 (1285)	660–18,850 (1275)	^b^ 0.963
	1395.0 ± 590.6	2412.5 ± 4414.7	

^b^ Mann–Whitney U Test.

## Data Availability

Data will be shared upon request.

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
