# Peer review of "Diagnosis of Latent Tuberculosis Infection in Hemodialysis Patients: TST versus T-SPOT.TB"

_diagnostics, 2023, doi:10.3390/diagnostics13142369_

Round 1
Reviewer 1 Report
I don't have any special suggestions, I think it's a good original
Author Response
Dear Reviewer,
Thank you very much for your evaluation.
Best regards,
Reviewer 2 Report
Binay et al. present a nice study of Tspot vs TST diagnostic testing for latent TB in a cohort of HD patients, comparing hematological and comorbidity profiles of Tspot pos vs neg patients, as well as a limited comparison of Tspot and TST outocomes for those patients who had both available.
The study is well set up, with useful and clinically/scientifically sound patient parameters compared for the groups of patients. It could benefit from some improvements to data presentation:
1. For patients with prior history of TB, please specify if this refers to prior latent or active TB, or both, as well as completeness of treatment, if known. Also, what was the time interval from prior TB to Tspot/TST testing?
2. For the difference in age correlating with higher rate of pos Tspot results, is this expected to be related to differences in TB incidence in Turkey over time (i.e. the older patients grew up and lived in Turkey during a time when TB was more highly endemic)? - would be good to see this discussion and correlation with rates in the discussion section.
3. The statement regarding history of prior TB not influencing Tspot outcomes seems tenuous given that only 2 patients are reported to have had a history of prior TB. While it's a good description of the population to know, would suggest to omit formal comparison of this parameter.
4. For patients with prior history of exposure to active TB patients, what was the time interval from exposure to Tspot/TST testing?
5. Would suggest to include the schedule for BCG vaccination in Turkey (e.g. at what age(s) is it typically given? ) as part of the discussion of BCG influence on TST/Tspot results
6. Section 3.2 states "there were 20% cases encountered with active TB patients" - not sure whether this means that 20% patients with TST had prior active TB history, or that 20% of patients received TST testing at the time when they had active TB. Please clarify.
7. Table 5 - please include information on white blood cells and lymphocyte numbers for patients who had Neg and Pos Tspot (would be good to see this breakdown specifically for those patients who also had TST, in addition to the overall cohort of patients who had Tspot).
8. The last paragraph of Discussion states that the rate of latent TB infection in HD patients is higher than in the normal population. This specific conclusion actually can't be drawn from the study - no stats on latent TB rates in a "normal" Turkish population, who would be subject to regular testing, are provided for comparison.
9. Figure 1 seems redundant - would suggest to omit.
There are some minor grammatical/stylistic changes that the paper would benefit from:
1. Materials and Methods - 3rd inclusion criterion should be "Those without concurrent active TB diagnosis"
2. Results paragraph 2 - instead of "Having had tuberculosis rate previously" should be "There were 3.6% (n=2) patients with prior history of TB" (and clarified whether active or latent or both, as per previous comment)
3. Table 1 - "Hypertension" (rather than "Hypertansion", which is a typo)
4. Section 3.2 of results should be "Results of patients with known history of TST". Paragraph 2 of the same section should state "it was observed tht 20% (n=2) HAD type 2 DM, etc." - "had" was omitted
5. Discussion, last paragraph on p. 7, second sentence, should read "... was higher in HD patients who had history of TB" (not "who were met with TB")
Author Response
Dear Reviewer,
Thank you very much for your valuable suggestions. In line with your suggestions, I made the necessary changes in the text and indicate it in red.
Point 1. For patients with prior history of TB, please specify if this refers to prior latent or active TB, or both, as well as completeness of treatment, if known. Also, what was the time interval from prior TB to Tspot/TST testing?
Response 1: Those previously treated for TB include treatment for active tuberculosis. There were no patients treated with LTBI. The patients stated that they completed their treatment. However, the time interval from the previous TB to the T-SPOT.TB/TST test could not be determined. Thank you very much for your valuable suggestions.
Point 2. For the difference in age correlating with higher rate of pos Tspot results, is this expected to be related to differences in TB incidence in Turkey over time (i.e. the older patients grew up and lived in Turkey during a time when TB was more highly endemic)? - would be good to see this discussion and correlation with rates in the discussion section.
Response 2: ‘The incidence of TB in our country has been decreasing over the years. So that, the incidence of TB, which was 29.8% in 2005, decreased to 14.4% in 2018 [32]. This may be the reason why T-SPOT.TB test gives high positive results in older ages.’ statement has been added to the relevant place. Thank you very much for your valuable suggestions.
Point 3. The statement regarding history of prior TB not influencing Tspot outcomes seems tenuous given that only 2 patients are reported to have had a history of prior TB. While it's a good description of the population to know, would suggest to omit formal comparison of this parameter.
Response 3: Based on your suggestion, this comparison has been cancelled. Thank you very much for your valuable suggestions.
Point 4. For patients with prior history of exposure to active TB patients, what was the time interval from exposure to Tspot/TST testing?
Response 4: The patients stated that the exposure was old, but they did not specify the exact duration. Thank you very much for your valuable suggestions.
Point 5. Would suggest to include the schedule for BCG vaccination in Turkey (e.g. at what age(s) is it typically given? ) as part of the discussion of BCG influence on TST/Tspot results
Response 5: ‘BCG vaccine is in the routine childhood vaccination schedule in our country and is administered at the end of the 2nd month [2]. It is important for our country that the T-SPOT.TB test is not affected by the BCG vaccine.’ statement has been added to the relevant place. Thank you very much for your valuable suggestions.
Point 6. Section 3.2 states "there were 20% cases encountered with active TB patients" - not sure whether this means that 20% patients with TST had prior active TB history, or that 20% of patients received TST testing at the time when they had active TB. Please clarify.
Response 6: "There were 20% cases encountered with active TB patients' was replaced with '20% (n=4) of the patients who underwent TST stated that they had encountered with active tuberculosis patient before.' Thank you very much for your valuable suggestions.
Point 7. Table 5 - please include information on white blood cells and lymphocyte numbers for patients who had Neg and Pos Tspot (would be good to see this breakdown specifically for those patients who also had TST, in addition to the overall cohort of patients who had Tspot).
Response 7: Table 5 is now Table 6 and the required analysis is included in Table 6. Thank you very much for your valuable suggestions.
Point 8. The last paragraph of Discussion states that the rate of latent TB infection in HD patients is higher than in the normal population. This specific conclusion actually can't be drawn from the study - no stats on latent TB rates in a "normal" Turkish population, who would be subject to regular testing, are provided for comparison.
Response 8: ‘the rate of latent TB infection in HD patients is higher than in the normal population.’ statement has been removed. Thank you very much for your valuable suggestions.
Point 9. Figure 1 seems redundant - would suggest to omit.
Response 9: Figure 1 has been removed. Thank you very much for your valuable suggestions.
In addition to the revisions, the grammatical corrections you suggested were also made. Thank you very much for your valuable suggestions again.
Best regards,

Reviewer 3 Report
The authors conducted a study with the aim of determining the frequency of latent tuberculosis (TB) infection among patients undergoing hemodialysis at Erzincan Binali Yıldırım University Hospital. Additionally, they sought to compare the effectiveness of different diagnostic tests for latent TB infection.
While the study objectives are intriguing, there are a few aspects that could benefit from improvement. One notable limitation is the relatively small sample size, which may affect the generalizability of the findings. To enhance the robustness of the study, a larger sample size could be considered in future research.
Furthermore, the statistical analysis employed in the study is primarily limited to univariate analysis. Although this analysis provides some insights, it does not allow for drawing strong conclusions or establishing comprehensive relationships between variables. A more sophisticated statistical approach, such as multivariate analysis, could be utilized to better explore the potential associations and interactions between various factors related to latent TB infection.
To enhance the overall quality of the paper, the introduction section could be further developed. This could involve providing a more comprehensive background on the significance of latent TB infection in the context of patients undergoing hemodialysis. Additionally, it would be beneficial to include a thorough review of the existing literature on the diagnostic tests used for latent TB infection, highlighting their strengths, limitations, and gaps in knowledge. By expanding on these aspects, the introduction section would offer a more comprehensive foundation for the study and demonstrate a deeper understanding of the research context.
The M&M section should be divided into many subsections (e.g., Study Design and Population, Methodology, Statistical Analysis).
Figure 1 should be removed.
The reference section should be reviewed.
Moderate editing of English language required.
Author Response
Dear Reviewer,
Thank you very much for your valuable suggestions. In line with your suggestions, I made the necessary changes in the text and indicate it in blue.
Point 1. Furthermore, the statistical analysis employed in the study is primarily limited to univariate analysis. Although this analysis provides some insights, it does not allow for drawing strong conclusions or establishing comprehensive relationships between variables. A more sophisticated statistical approach, such as multivariate analysis, could be utilized to better explore the potential associations and interactions between various factors related to latent TB infection.
Reponse 1. Multivariate analysis was performed. ‘When we evaluated the risk factors affecting the T-SPOT.TB test such as age, gender, previous TB finding on pa chest X-ray, time of dialysis and encountering an active tuberculosis patient with Enter Logistic regression analysis; the model was found to be significant and the explanatory coefficient of the model (76.8%) was found to be at a good level. It is seen that the effect of a unit increase in age on T-SPOT.TB positivity increases the ODDS ratio 1.101 (95% CI: 1.016-1.192) times. The effect of encountering an active tuberculosis patient has an effect on T-SPOT.TB positivity with an ODDS value of 59.762 (95% CI:1.59-2233.42) times. The effects of gender, previous TB finding on pa chest X-ray and time of dialysis were not significant in multivariate evaluation (p>0.05) (Table 4).’ statement has been added. Thank you very much for your valuable suggestions.
Point 2. To enhance the overall quality of the paper, the introduction section could be further developed. This could involve providing a more comprehensive background on the significance of latent TB infection in the context of patients undergoing hemodialysis. Additionally, it would be beneficial to include a thorough review of the existing literature on the diagnostic tests used for latent TB infection, highlighting their strengths, limitations, and gaps in knowledge. By expanding on these aspects, the introduction section would offer a more comprehensive foundation for the study and demonstrate a deeper understanding of the research context.
Reponse 2. ‘When the studies are examined; In the systematic reviews conducted by Alemu et al., LTBI and active tuberculosis infection were found to be more common in dialysis patients [3,4]. In the study of Xia et al., it was found that the rate of development of active tuberculosis was higher in hemodialysis patients with LTBI. In the same study in which patients were followed up about 3 years, LTBI was also shown to be associated with major adverse cardiovascular events [5]. In the study of Park et al., it was shown that active tuberculosis is more common in dialysis patients and kidney transplant recipients compared to the general population and causes higher mortality rates [6]. In the study of Romanowski et al., it was found that active tuberculosis was seen less frequently in the patients which were treated for LTBI [7].
Interferon Gamma Release Assays (IGRA) and Tuberculin Skin Test (TST) are used in LTBI screening, and it is recommended to perform IGRA in immunocompromised groups such as hemodialysis patients when TST is negative or cannot be performed. Among the IGRA tests, T-SPOT.TB or Quantiferon-TB Gold In Tube (QFT-GIT) test is used [2,8-12]. When the studies comparing the diagnostics tests are examined, there is no gold standard test. In the study of Akbar et al., QuantiFERON-TB gold plus test was shown to be su-perior to TST. However, the small sample size was determined as a limitation of the study [13]. On the other hand, in the study of Setyawati et al., it was recommended to use the TST in the diagnosis of LTBI [14]. However, although it is stated that IGRA tests are not affected by immunosuppression, studies in patients with chronic kidney disease have shown that as the duration of dialysis increases, IGRA tests are more likely to give false-negative results. In this context, it is recommended that patients with chronic kidney disease be screened for LTBI at an early stage [15-18]. Considering the systematic reviews carried out in recent years, it has been shown that IGRA tests are superior to TST [11, 19, 20].’ statement has been added to ‘INTRODUCTION’ section. Thank you very much for your valuable suggestions.
Point 3. The M&M section should be divided into many subsections (e.g., Study Design and Population, Methodology, Statistical Analysis).
Reponse 3. The M&M section has been divided into Study Design and Population, Methodology, Statistical Analysis. Thank you very much for your valuable suggestions.
Point 4. Figure 1 should be removed.
Reponse 4. Figure 1 has been removed. Thank you very much for your valuable suggestions.
Point 5. The reference section should be reviewed.
Reponse 5. The reference section has been reviewed. Thank you very much for your valuable suggestions.
Thank you very much for your valuable suggestions again.
Best regards,

Round 2
Reviewer 3 Report
After carefully reviewing the authors' modifications, I recommend accepting the manuscript in its current form. The necessary revisions have been adequately addressed, and the manuscript now meets the requirements for publication.
NA